# Estrogen Receptors as Molecular Targets of Endocrine Therapy for Glioblastoma

**DOI:** 10.3390/ijms222212404

**Published:** 2021-11-17

**Authors:** Andrea Magali González-Mora, Patricia Garcia-Lopez

**Affiliations:** 1Laboratorio de Farmacologia, Subdirección de Investigación Básica, Instituto Nacional de Cancerología, Ciudad de México 14080, Mexico; maga_mor@ciencias.unam.mx; 2Facultad de Ciencias, Universidad Nacional Autónoma de México (UNAM), Ciudad de México 04510, Mexico

**Keywords:** glioblastoma, endocrine therapy, estrogen receptors, drugs, resistance to chemotherapy

## Abstract

Hormonal factors may participate in the development and progression of glioblastoma, the most aggressive primary tumor of the central nervous system. Many studies have been conducted on the possible involvement of estrogen receptors (ERs) in gliomas. Since there is a tendency for a reduced expression of ERs as the degree of malignancy of such tumors increases, it is important to understand the role of these receptors in the progression and treatment of this disease. ERs belong to the family of nuclear receptors, although they can also be in the plasmatic membrane, cytoplasm and mitochondria. They are classified as estrogen receptors alpha and beta (ER⍺ and ERβ), each with different isoforms that have a distinct function in the organism. ERs regulate multiple physiological and pathological processes through the activation of genomic and nongenomic pathways in the cell. Nevertheless, the role of each isoform in the development and progression of glioblastoma is not completely clear. Diverse in vitro and in vivo studies have shown encouraging results for endocrine therapy as a treatment for gliomas. At the same time, many questions have arisen concerning the nature of ERs as well as the mechanism of action of the proposed drugs. Hence, the aim of the current review is to describe the drugs that could possibly be utilized in endocrine therapy for the treatment of high-grade gliomas, analyze their interaction with ERs, and explore the involvement of these drugs and receptors in resistance to standard chemotherapy.

## 1. Introduction

Estrogens are lipophilic hormones with a steroid origin and have important effects on the reproductive, digestive, immune, cardiovascular, endocrine, respiratory and central nervous systems [1,2,3]. Physiologically, they participate in diverse cellular processes, such as electrolytic balance, the metabolism of lipids and carbohydrates, the homeostasis of osteocytes, and different functions in the skin [2]. 

In premenopausal women, estrogen production mainly takes place in the ovaries. In men and postmenopausal women, the principal pathway for the generation of estrogens is extragonadal [4]. There are four types of estrogens: 17β-estradiol (E2, the most abundant), estrone (E1), estriol (E3) and estetrol (E4). Two androgens, testosterone and androstenedione, are converted locally into E2 by aromatase (an enzyme of the superfamily of cytochrome P450) at various sites, including the breasts, brain and adipose tissue [5]. 

Estrogens trigger signaling activity by binding to estrogen receptor alpha (ER⍺) and estrogen receptor beta (ERβ). Although these estrogen receptors (ERs) belong to a large family of nuclear receptors, they are also found in the cytoplasm, the mitochondrion and the cell membrane [4,5,6]. ERα and ERβ are routed to the plasmatic membrane by a post-translational modification called palmitoylation. These two types of ERs have different isoforms, which result from the alternative splicing in the genes that encode them [5] (Figure 1). ER⍺ and Erβ are encoded in *ESR1* (chromosome 6q25.1) and *ESR2* (chromosome 14q23.2), respectively, through a process of genetic duplication [7].

The diverse mechanisms of action of ERs can be grouped into genomic and nongenomic pathways (Figure 2). The former consists of the regulation of genetic expression through the activation of ERs by E2 or by phosphorylation, followed by the recognition of the promoter region in the target genes, known as the estrogen response element (ERE). The consensus sequence of ERE is AGGTCAnnnTGACCT, where “n” is any nucleotide. About one third of the genes regulated by ERs do not contain the ERE sequence. In such cases, activity is mediated by the interaction of ERs with other transcriptional factors capable of recognizing specific binding sites [1,4].

The nongenomic pathways are those in which the receptors do not act as a transcriptional factor. They induce effects at a faster speed than the genomic pathways, regulating signaling by interacting with scaffolding proteins or with other proteins related to the plasmatic membrane. Nongenomic activity may indirectly lead to the activation or repression of genes [1]. The genomic and nongenomic pathways can be coupled to promote the transcriptional activation and/or repression of numerous genes to regulate the physiological state of cells [4]. 

Membrane estrogen receptors (mERs) participate in the activation of signaling pathways; the most recognized being the mitogen-activated protein kinase (MAPK) pathway also known as the extracellular signal-regulated kinase (ERK), or MAPK/ERK pathway. Through rapid responses to external stimuli, mERs can initiate the transcription of diverse genes regulated by estrogens. They are located mainly in invaginations of the membrane denominated caveolae, comprised of caveolin 1 and caveolin 2, which can act as scaffolds for signaling molecules [7,8,9,10,11,12]. 

Estrogen also activates the G protein-coupled ER (GPER1 or GPR30), which is involved in nongenomic signaling pathways. It is encoded by the *GPER1* gene, located in chromosome 7p22.3. In 1997, the presence of this gene was identified in MCF-7 and MDA-MB-321 cells of breast cancer and in tissues from patients with breast cancer. Additionally, GPER1 is expressed in the tissues of different organs, including the heart, brain, lungs, liver, skeletal muscle, kidneys, pancreas, and especially in the placenta [13,14,15]. It plays a role in the regulation of multiple signaling pathways, such as cyclic adenosine 3′,5′-monophosphate (cAMP), Notch, the epidermal growth factor receptor (EGFR), the insulin-like growth factor receptor (IGFR), nuclear factor kappa light chain enhancer of activated B cells (NF-κB), the Hippo/Yes-associated protein (Hippo/YAP), phosphatidylinositol 3-kinase (PI3K)/protein kinase B (AKT), and MAPK/ERK. Under conditions of hypoxia, GPER1 interacts with the hypoxia inducible factor (HIF)-1α to activate genes that code for the connective tissue growth factor (CTGF), the vascular endothelial growth factor (VEGF) and interleukin (IL)-6, among others. Given its broad participation in cellular dynamics, GPER1 is implicated in the development and progression of different types of cancer, including breast, endometrial, prostrate, ovarian, cervical, and lung cancer [13,16,17].

## 2. Estrogen Receptors in Cancer 

ERs can regulate diverse physiological processes at the cellular and systemic level. However, they have also been linked to the progression of various types of cancer, such as breast, ovarian and endometrial cancer in women [18]. In men, estrogens may play a key role in the progression of prostate cancer [19]. Moreover, estrogens are thought to have a protective function in the colon, liver, and other organs [20,21,22]. 

Although the participation of ERs in the development and progression of different types of cancer is still not completely clear [18], much has been learned through intense research on their impact in some types of cancer, especially breast cancer. The subtypes of the latter disease are defined by the presence or absence of certain molecular markers, a classification that has allowed for a more accurate prognosis for breast cancer patients. For instance, patients with luminal A or B tumors have a better prognosis of survival than patients with triple negative tumors. In luminal A and B tumors, ERs and progesterone receptors (PRs) are highly expressed, and the human epidermal growth factor receptor (HER2) is not expressed. In triple negative tumors, ERs, PRs and HER2 are absent [23,24]. The pharmacological treatment for hormone-dependent tumors is based on endocrine therapy [25,26]. 

## 3. Endocrine Therapy

Endocrine therapy impedes the proliferation of cancer cells by using antagonists for hormone receptors or inhibiting hormone production. There are four main types of molecules commonly administered in this type of cancer therapy: (1) selective ER modulators (SERMs), (2) selective ER degraders (SERDs), (3) aromatase inhibitors, and (4) agonists of the luteinizing hormone-releasing hormone (LHRH) [25].

### 3.1. Selective Estrogen Receptor Modulators (SERMs)

SERMs promote a conformational change that imitates the activated state of ERs, while at the same time preventing the binding of coactivators. The resulting inactivation of ERs blocks their effects on cells. SERMs include tamoxifen, raloxifene, toremifene, arzoxifene, bazedoxifene, pipendoxifene and lasofoxifene [26]. 

These compounds are principally employed to treat breast cancer positive to hormone receptors. Since their effect is tissue specific, other applications also exist, such as prophylaxis against osteopenia and osteoporosis [26,27]. 

Tamoxifen is the most widely used prodrug in the treatment of breast cancer positive to hormones, especially in premenopausal women. Its active metabolite, 4-hydroxytamoxifen, impedes cell proliferation. Through agonist or partial agonist activity, it favors the proliferation of endometrial cells when administered for prolonged periods of time, which may lead to endometrial cancer [25]. Although tamoxifen, toremifene, and raloxifene have all demonstrated similar positive outcomes in the treatment of breast cancer, the latter two do not stimulate cell growth in the uterus. However, raloxifene produces more severe adverse effects than tamoxifen and toremifene [28,29,30,31]. 

### 3.2. Selective Estrogen Receptor Degraders (SERDs) 

Upon coupling to the ligand binding site of the receptor, SERDs destabilize ERs by evoking a conformational change that modifies the activity of both the activation function (AF)-1 and AF-2 domains. In contrast, tamoxifen only alters the AF-2 domain [32,33]. Some SERDs utilized in endocrine therapy are ICI 182780, GW5638, Gw7604, elacestrant, GDC0810 and AZD9496 [26]. ICI 182780, the most thoroughly studied SERD in cancer therapy, evokes a conformational change that induces the polyubiquitination of the receptor, a signal for the activation of the proteasomal degradation pathway. Hence, ICI 182780 not only diminishes the signaling of ERs but also decreases their presence in tissue. In this way, the dimerization and nuclear localization of these receptors is avoided, which in turn limits the transcription of genes regulated by them. ICI 182780 is considered a pure antagonist, based on its agonist activity of only 5%. In contrast, tamoxifen is classified as a partial antagonist because it exhibits an agonist activity of 23–30% [32,33,34].

### 3.3. Aromatase Inhibitors

Aromatase is an enzyme responsible for the synthesis of estrogens from androgens. Aromatase inhibitors, such as anastrozole, letrozole, formestane, aminoglutethimide and exemestane, limit the aromatization of androgens and thus reduce (up to 99%) the levels of estrogens, which can reach undetectable levels in circulation. The use of aromatase inhibitors is mainly indicated for postmenopausal women. In premenopausal women, the peripheral depletion of the level of estrogens is a positive signal to produce estrogens in the ovaries [25,35].

### 3.4. Agonists of the Luteinizing Hormone-Releasing Hormone (LHRH)

By binding to its receptors in the pituitary gland, LHRH mediates the release of the luteinizing hormone (LH) and follicle-stimulating hormone (FSH), two gonadotropic hormones. Therefore, it also denominates the gonadotropin-releasing hormone. The luteinizing hormone and follicle-stimulating hormone are responsible for promoting the steroidogenesis of certain hormones, including androgens [36,37].

The binding of LHRH agonists to their receptors generates LH and FSH during the first few days. Subsequently, aggregates of the ligand-receptor complex are formed and internalized by the cells, which diminishes the number of receptors on the cell surface. By 21 days after administration of the agonist, the lower number of LHRH receptors has reduced the presence of the LH and FSH, which reach levels similar to those of postmenopausal women. Consequently, the hypothalamic−pituitary−gonadal axis is inhibited, and the synthesis of steroid hormones decreases [25].

## 4. Estrogen Receptors in Glioblastoma

Glioblastoma is the most common and most aggressive primary brain tumor. According to the fifth edition of the WHO Classification of Tumors of the Central Nervous System (WHO CNS5), glioblastoma is classified using histopathological and molecular criteria. It considers histological findings, as a high proliferation rate, microvessel formation, necrosis, and the diagnostic complements with the identification of biomarkers as IDH-wildtype, TERT promoter mutation, +7/−10 chromosomes copy number changes and EGFR gene amplification. In the CNS5, it is reported that glioblastoma belongs to the “Adult-type diffuse gliomas” family, with a grade 4 of malignancy that has a poor survival rate and the lack of an effective therapy [38]. Based on the phase 3 EF-14 clinical trial, progression-free survival (PFS) has increased from 4.7 months to 6.7 months whit the addition of tumor-treating fields (TTFields) therapy to standard chemotherapy with temozolomide; while overall survival (OS) has improved to 20.9 months compared to 16 months with temozolomide alone, and the 5-year-survival rate was 13% versus 5%, [39]; unfortunately, the patients with glioblastoma have a 100% relapse rate.

Although the risk factors associated with this disease are not completely defined, age and some other components are known to play a role. Glioblastoma is diagnosed at an average age of 65 years [40]. Less than 1% of glioblastoma tumors are linked to hereditary cancer syndromes (e.g., neurofibromatosis 1 and 2, the Turcot syndrome, or the Li-Fraumeni syndrome). Ionizing radiation is also a factor correlated with a high risk for the disease. There is a greater incidence in men versus women (1.6:1), suggesting a hormonal influence during the development and progression of the disease. Based on epidemiological data, women have a better response to treatment than men, indicating that estrogens could possibly have a protective function [41,42,43,44].

The role played by estrogen and its receptors during the development of glioblastoma is still not clear. Due to the cellular characteristics of this neoplasm, glial cells have long been assumed to represent its origin. The expression of ER⍺ is lower in glia than low-grade gliomas, and as the degree of malignancy of the glioma increases, such expression decreases. In contrast, the expression of ERβ is greater in glia than gliomas. On the other hand, patient samples show no difference in the expression of GPER1 between these two tissues. However, in this study the size of the corresponding sample may not be representative, as it included only four patients with glioma grade III, three patients with glioblastoma, and four patients without any tumor [45,46]. The possible regulatory role of GPER1 during the progression of gliomas should be explored in future research.

According to a recent report, astrocytes, oligodendrocytes and other totally differentiated cells are unlikely to undergo a malignant transformation. The more differentiated, the less susceptible cells are to malignant transformation. When Alcantara-Llaguno et al. (2019) suppressed genes important in glioblastoma (*Trp53*, *Pten* and *Nf1*) in a murine transgenic model, a malignant phenotype was generated in certain types of cells, such as neural stem cells (NSC), bipotential progenitor cells, and oligodendrocyte progenitor cells. However, this effect was not observed in neural progenitor cells in the late stage of development, in newborn neurons, or in postmitotic neurons [47,48]. Interestingly, ERα and ERβ are expressed in embryonic and adult NSC extracted from rats. A relatively high expression of ERα is found in NSC during the E15–E17 stages of embryonic development, while a reduced expression exists in adults. The expression of ERβ, on the hand, remains constant during development and is elevated in the NSC of adults [49]. Although the uncertainty about the cellular origin of gliomas and the role of ERs during the malignant transformation, relative low ER expression in tumors is clearly associated with a higher degree of malignancy. A tumor-suppressing function is frequently proposed for ERβ and a tumor-promoting function for ER⍺ [50,51,52].

Despite the multiple studies carried out to date, the role of ERs and E2 in gliomas is not completely clear. Hönikl et al. (2020) contradicts the previously established data, by proposing that in glioblastoma, both male and female patients have an improved probability of longer survival if they exhibit an elevated expression of ER⍺ and aromatase, implying the capacity of both to suppress tumors. The authors collected tissue samples from patients and classified them according to low (<40%) and high (>40%) ER expression. Patients with a high expression had a longer survival time [53].

In the same study, LN18 and LN229 cells of glioblastoma (positive to ER⍺) were exposed in vitro to increasing concentrations of E2 (from 10–50 μM), resulting in a dose-dependent decrease in viability. When applying E2 prior to temozolomide, there was a greater sensitivity to treatment with the latter in the LN229 cell line [53], demonstrating the important role of ER⍺ and aromatase in gliomas. It is not known whether the favorable outcome of estrogen treatment could cause adverse effects, which would be more likely in male patients [54]. Another relevant issue is whether estrogen-based hormone therapy would be beneficial only for patients with a high expression ER⍺, or also for those with a low or null expression.

Unlike the findings described by Hönikl et al. (2020), the in silico evaluation conducted by Hernández-Vega et al. (2020) revealed a lower probability of survival for patients with an elevated expression of ER⍺ and ERβ. The information for the analysis was obtained from the Cancer Genome Cancer Atlas and the GTEx database. The patients with a reduced expression of these receptors showed better survival in spite of having a robust expression of the mesenchymal subtype, considered to be the subtype with the worst prognosis in glioblastoma [55]. The differences in the results might be due to the size of the sample or to the region of the brain evaluated. Further research is necessary to gain insights into the mechanisms associated with ER⍺ and ERβ, since they are clearly involved in the development and treatment of glioblastoma.

Although diverse studies have been carried out to determine the expression of ERs in gliomas, little evidence exists about the relevance of ER isoforms as possible therapeutic targets. Regarding ER⍺, the ERα36 isoform is expressed in tumors from patients and in cell lines of glioblastoma, including U251 and U87-MG. Found in the cytoplasm and plasmatic membrane and anchored by caveolin-1, ERα36 has been linked to resistance to tamoxifen, perhaps mediated by a positive regulation of autophagy [56,57].

Concerning ERβ isoforms, it has been reported that an increase in the expression of isoforms ERβ2 and ERβ5 have a prognostic significance in prostate and ovarian cancer, being associated to a poor prognosis in both cancers [58,59]. However, regarding glioma, previous findings suggest that ERβ5 can play a protective role in glioblastoma. Li et al. (2013) report that ERβ5 is highly expressed in glioma compared with non-neoplastic brain tissue and this expression is increased by hypoxic conditions, promoting an inhibition of PI3K/AKT/mTOR pathway through the induction of PTEN. It was also demonstrated that in the U87 cell line transfected to over-express ERβ1 and ERβ5, there was a significant reduction in cellular proliferation [60].

On the other hand, Liu et al. (2018) evaluated the role of distinct isoforms in glioblastoma (e.g., ERβ1, ERβ2, ERβ4 and ERβ5) in tissues from patients as well as in cell lines and primary cultures of brain tumors. ERβ5 was more abundant in high-grade brain tumor tissue and in the cell lines of glioblastoma than in tissue from healthy individuals and from low-grade glioma tumors. When examining the relation of each isoform to viability, proliferation, invasive capacity, migration, and colony-forming capacity, ERβ1 proved to have a tumor suppressor function and ERβ5 showed oncogenic properties. According to the analysis of the RNA sequence, ERβ1 modulates genes negatively correlated with the signaling pathways of NF-κB and Janus kinase signal transducer and activator of transcription proteins (JAK-STAT), while ERβ5 regulates genes positively correlated with the same signaling pathways. In relation to the protein−protein interactions of both ERβ1 and ERβ5, only the latter interacts with proteins involved in the regulation of the signaling pathways of the mammalian target of rapamycin (mTOR), DNA repair, and proteins linked to migration and invasion pathways, such as T-complex 1α (TCP-1α), four and a half LIM domains protein (FHL)-2, and filamins. In their orthotopic mouse model of implanted glioma cells, animals survived longer after being infected with tumor cells expressing only ERβ1 versus those expressing ERβ5 as well. The authors concluded that the activation of ERβ1 and the attenuation of the activity of ERβ5 could serve as an alternative therapy for glioblastoma [61].

Although the work of Liu et al. (2018) is experimentally more robust, this will help to determine if the discrepancy between the two articles could be due to technical issues related to specific antibodies, experimental conditions, or if it is due to some biological issue that is being overlooked.

On the other hand, the difference in the function of ERβ1 and ERβ5 isoforms may be explained by their distinct structures. ERβ1 is a full-length isoform in which the carboxyl terminal domain contains helix 12, essential for generating the binding cavity for the ligand (in the AF-2 domain). Moreover, it can form homodimers and bind to estrogens. Contrarily, ERβ5 has a stop codon in exon 8, giving rise to a truncated helix 11 and the absence of helix 12. This structure impedes the formation of homodimers. Hence, ERβ5 has a ligand-dependent genomic activity when acting as a heterodimer with ERβ1. The heterodimers formed by the combination of ERβ1 and ERβ5 recruit one set of coregulators and the heterodimers of ERβ1 alone recruit another set, each with a distinct transcriptional activity. If the expression of ERβ1 and ERβ5 is modified, their interaction is probably altered as well, originating a physiological imbalance in the tissues [62].

## 5. Endocrine Therapy for Glioblastoma

The standard therapy for glioblastoma consists of surgical resection followed by radiation and chemotherapy with the alkylating agent temozolomide [63]. A meta-analysis was published of the survival of patients, pre- and post-2005, under this treatment schema. The results showed a doubling of survival rates at 2 and 3 years after diagnosis in older patients (>65 years), as from 2005, although the 5-year-survival rate is still showing poor improvement [64]. This indicates that new therapeutic options are necessary. One of the problems in the search for novel therapies is the scarcity of drugs capable of crossing the blood−brain barrier (BBB) [65]. Within the broad spectrum of drugs used in endocrine therapy (mainly for breast cancer), some compounds have been selected to test their therapeutic effect on gliomas, especially glioblastoma.

### 5.1. Selective Estrogen Receptor Modulators (SERMs)

Tamoxifen is the most studied drug as a possible endocrine treatment and sensitizer to chemotherapy for gliomas, resulting in multiple publications on the plausible molecular mechanisms of this agonist/antagonist in malignant cells. Pollack et al. (1990) was the first group to evaluate tamoxifen as an antineoplastic agent for gliomas, finding it to have an antiproliferative effect. Additionally, they demonstrated that the signaling pathway mediated by protein kinase C (PKC) regulates the mitogenic process in cells. According to the same group (1995), tamoxifen produces a dose-dependent inhibition of the proliferation of cell lines from low- and high-grade glioma tumors in pediatric patients [66,67]. In 2014, they confirmed that the inhibition of the PKC pathway is one of the principal mechanisms by which tamoxifen inhibits cell proliferation and migration, observing a tamoxifen-induced increase in apoptosis and cell cycle arrest. The effect was potentiated by combining tamoxifen with temozolomide [68].

Other studies have revealed that the antineoplastic activity of tamoxifen is not only generated through the PKC pathway, but also by the activation of the c-Jun N-terminal kinase (JNK)-1 and caspase-3 pathways, leading to a low expression of Bcl-2 [69]. Moreover, tamoxifen gives rise to cell death in glioblastoma tumors independently of the caspase pathway. The corresponding cytotoxicity may be linked to greater autophagy, mainly in the U87 cells with a wild-type EGFR. Hence, certain molecular profiles are likely to be more susceptibility to treatment [70].

Although tamoxifen has been proven to have important activity against the progression of gliomas, there are mechanisms of resistance that can limit the efficacy of this treatment. For example, when tamoxifen binds to ER⍺36, highly expressed in glioblastoma, the activation of this receptor contributes to the enhancement of autophagy detected in U87 and U251 cells, thus causing resistance to apoptosis. The silencing of ER⍺36 reduces autophagy and diminishes cell viability [56].

Increased autophagy as a mechanism of resistance to treatment is controversial. Autophagy induced by tamoxifen is described to be cytotoxic for glioblastoma (Graham et al., 2016). However, according to the report by Qu et al. (2019), tamoxifen increases autophagy and at the same time boosts the levels of ERα36 and cancer cell viability. Thereby, ERα36 promotes resistance as a result of the agonist effect of tamoxifen. Furthermore, Liu et al. (2016) pointed out that the apoptosis elicited by tamoxifen through the activation of the caspase pathway and the regulation of survivin was reversed by ICI 162,486, a selective agonist of ERα36 [56,57,70].

Even though there exist several in vitro assays probing the effectivity of cytotoxic tamoxifen effect using glioblastoma cell cultures, clinical trials have shown discordant results in therapeutic improvement [71,72,73,74,75,76].

In order to test the safety and efficacy of tamoxifen in the treatment of recurrent glial tumors, Couldwell et al. (1996) administered high doses of tamoxifen (200 mg/m^2^ for men and 160 mg/m^2^ for women twice daily). Among the 20 patients with glioblastoma treated, a mean survival of 17.4 months was achieved. Previous nitrosourea therapy failed in six of the participating patients, who showed stable disease or response to treatment with tamoxifen. Although during the study, some of the patients experienced adverse effects such as deep vein thrombosis events, nausea and hot flashes. However, the author concludes that most patients tolerated the therapy well [71]

In the same way, Chang et al. (2004) reported in a phase II study, the use of high doses of tamoxifen (escalated from 120 to 240 mg/day for one week) combined with oral etoposide (50 mg/m^2^/day for three weeks) in patients with recurrent glioblastoma. The results indicate that tamoxifen has limited toxicity and was well tolerated by patients. Although the efficacy was limited, probably due to highly resistant populations that prevent the synergistic effect of the combination [72].

In another phase II study, Robins et al. (2006) reported the use of high doses of tamoxifen (80 mg/m^2^) orally daily divided into doses was administered after conventional radiotherapy (30 fractions of 2 g). In this study, the difference was not significant between the control group (historical control taken from a previous glioblastoma study), and that of patients newly diagnosed with GBM. Interestingly, in this study it was shown that patients who received tamoxifen therapy had a reduction in about 10% of thromboembolic events typical of patients with brain tumors. This result implies a benefit for these patients. Likewise, this study proposes the use of other drugs such as temozolomide in combination with tamoxifen, due to its low toxicity [73].

In 2012, Patel et al. published a phase I study in which it was mentioned that 17 patients, recently diagnosed with a high-grade glioma, were treated with different regimens of temozolomide, radiation and tamoxifen. The objective was to determine the maximum tolerated dose (MTD) with this combined treatment. It was determinate that 100 mg/m^2^ of tamoxifen per day as the MTD when it is combined with 75 mg/m^2^ of TMZ and radiation (60 Gy divided into 2 Gy/day). The mean survival of this study was 17 months. Four glioblastoma patients survived more than 30 months and 35% survived more than 2 years. These results indicate that the combination of temozolomide with radiation and tamoxifen are favorable with respect to the survival offered by the concomitant therapy of temozolomide and radiation (14.6 months mean survival and overall survival at 2 years of 26.5% of patients) [74].

Moreover, in the study of DI Cristofori et al. (2013), dense doses of temozolomide (75–150 mg/m^2^/week with one week off), combined with daily doses of tamoxifen (80 mg/m^2^), were administered in patients with recurrent glioblastoma. As in previous studies, the therapy was well tolerated. It was also effective by increasing the efficacy of temozolomide, having an OS (overall survival) of 17.5 months and a TTP (time to recurrence) of 7 months [75]. However, in the retrospective study presented in 2020 by De Roxas et al., 167 patients with recurrent high-grade glioma were included. It was observed that there was no significant difference in mean overall survival and time to progression between the patients treated with tamoxifen (100 mg twice daily), and control group (vehicle). Even so, a subpopulation of patients was found to show stable disease or partial response to tamoxifen, showing improved survival. However, this study had limitations due to its nature, for which the author proposed improvements [76].

Even though tamoxifen is the most studied antiestrogenic drug in clinical study phases in glioblastoma, a consensus has not been reached about the effectiveness of this treatment in high-grade glioma patients, specifically in glioblastoma. According to some of the studies reported here, high doses of tamoxifen offer a positive effect in subpopulations or even promote an improvement in the mean survival of patients when combined with standard therapy. Nevertheless, other publications show a non-significant effect with respect to the control. This implies that it is necessary to consider the existing preclinical and clinical trials for further research with this drug to determine the appropriate experimental conditions and inclusion criteria to obtain conclusive results.

Other types of SERM have also been studied apart from tamoxifen, including raloxifene and bazedoxifene. Baritchii et al. (2016) evaluated the combination of different drugs, such as tamoxifen or raloxifene plus temozolomide, applied together with radiation, to treat cancerous stem cells (CSC) and the primary cultures of tissues from patients with glioblastoma. In individual treatments, tamoxifen and raloxifene have shown substantial inhibition of the cell viability of CSC but not that of primary cultures. When combining each SERM with temozolomide, the inhibitory activity is potentiated in both CSC and primary cultures. When tamoxifen or raloxifene are applied in conjunction with radiation, there is a lower viability of CSC than that found with temozolomide plus radiation. On the other hand, the addition of temozolomide to SERM followed by radiation, did not potentiate the negative impact on the survival of CSC. Raloxifene displayed the same inhibitory activity as tamoxifen on the viability of these cells, an effect enhanced by the addition of temozolomide or radiation [77].

With an in vitro model, Fu et al. (2020) demonstrated the capacity of bazedoxifene to suppress the survival and invasiveness of glioma cells and foster apoptosis. This activity was potentiated with a combination of bazedoxifene plus paclitaxel. The combination proved capable of reducing tumor growth in an orthotopic model of glioblastoma. According to the researchers, a possible mechanism of action of bazedoxifene for suppressing tumor growth is through the activation of the Hippo/YAP signaling pathway [78].

### 5.2. Selective Estrogen Receptor Degraders (SERDs)

Among the SERDs tested to date, the best antineoplastic activity against gliomas has been exhibited by ICI 182780. An in silico study by Kotelnikova et al. (2010) revealed the capability of this antiestrogenic agent to inhibit signaling pathways relevant to halting the progression of the disease [79]. Even though Hui et al. (2004) did not observe any significant effect of the drug on the viability of U87, U138 and U373 cell lines of glioblastoma [80], the absence of effect may have been due to the short time of exposure to the treatment. Further research is needed to determine the efficacy of ICI 182780 on glioma cells.

Various mechanisms not related to hormone receptors have been described for ICI 182780. For instance, it can act as an inhibitor of ATP-binding cassette (ABC) transporters, such as P-glycoprotein (P-gp). The latter membrane protein functions as an efflux pump and is highly expressed in glioma tumors and in endothelial cells of the BBB. It participates in the pumping of chemotherapy drugs out of tumor cells, representing an important mechanism of resistance to chemotherapy treatment in diverse types of cancer, including glioblastoma [81].

Huang et al. (2017) showed that ICI 182780 and doxorubicin are substrates of P-gp. The competitive inhibition of P-gp by ICI 182780 reverses resistance to doxorubicin in Bats-72 and Bads-200, two cell lines of breast cancer negative to ERs. The combination of ICI 182780 with doxorubicin favors an elevated intracellular accumulation of the latter compound, with a slower efflux and a greater nuclear location. Thus, doxorubicin can interact more efficiently with the DNA of tumors to impede its replication. The combination treatment generates enhanced cytotoxicity, inducing apoptosis and promoting cell cycle arrest in the G2/M phase [82].

According to different studies, both temozolomide and ICI 182780 are transported by P-gp [83,84]. Hence, the combination of both treatments could contribute to the competitive inhibition of P-gp and the consequent improvement in the chemotherapeutic effect of temozolomide. ICI 182780 is able to cross the BBB [85], representing an advantage over many other agents to the treatment of glioblastoma.

### 5.3. Aromatase Inhibitors

Another promising area of research is the treatment of gliomas with aromatase inhibitors, some of which have already been tested. For example, anastrozole and letrozole decrease the viability and proliferation of C6 cells. In an orthotopic model, these two compounds diminish cell proliferation and tumor growth as well as the expression of aromatase and of GPER1. There is an increase in the expression of caspase 8 and 9, indicating greater apoptotic activity. Moreover, a higher expression of ER⍺ was observed, probably due to a compensatory activity resulting from the lower levels of E2 [86,87].

Letrozole has a cytotoxic effect on the C6 cell line of gliomas, manifested as decreased cell viability, and produces a decline in aromatase activity. In an in vivo model, this compound was found to reduce the growth rate of tumors in rats [88]. It was also assessed on primary cultures derived from patients with glioblastoma, being encapsulated with poly (lactic-co-glycolic acid) (PLGA) nanoparticles conjugated with anti-GD2 antibodies. GD2 is the specific target in glioblastoma tumor cells. Letrozole was proved to limit the proliferation and migration of tumor cells and the formation of spheroids. A lower number of tumor cells was observed with the combination treatment of letrozole plus temozolomide compared to the individual treatment with the former [89].

A phase 0/1 trial is currently being conducted in recurrent high-grade patients to assess the ability of letrozole, in combination with standard therapy, to penetrate BBB and concentrate in tumoral tissue. However, the results are not yet available [90]. Therefore, the usefulness of this drug in the treatment of patients cannot yet be determined.

Based on an in silico analysis of protein-ligand binding, exemestane might have a therapeutic effect by interacting with cell division protein kinase (CDK)-6, retinoblastoma protein (Rb), calcium/calmodulin-dependent protein kinase type II gamma chain (CAMK2G) and beta chain (CAMK2B), the N-ras protein, PKCβ, and the platelet-derived growth factor receptor (PDGFR)α, which are important proteins in the progression of glioblastoma. The authors suggested the possibility of adding the EGFR inhibitor gefitinib to the exemestane treatment to reduce the drug resistance and tumoral progression through the inhibition of EGFR translocation to the mitochondria [91].

### 5.4. Agonists of the Luteinizing Hormone-Releasing Hormone (LHRH)

The receptor for LHRH has been detected in the U87MG and U373 cell lines of glioblastoma, as well as in tumor tissue of the same cancer. In the presence of agonists of LHRH, there is a lower level of cAMP in glioblastoma tumor cells and a reduction in cell proliferation. The decline in the level of cAMP could indicate the existence of an interaction between the receptor of LHRH and the Gαi protein to mediate the antiproliferative effect [92].

It is noticeable that the use of aromatase inhibitors and the gonadotropin-releasing hormone has been proven to effectively inhibit the pathological processes of gliomas in preclinical studies. The function of these drugs is not to act directly on ERs, but rather to impede the synthesis of estrogens. This is relevant because E2 promotes the proliferation of the U373 and D54 cell lines of glioblastoma. Both cell lines express ER⍺, which can produce an effect through the genomic pathway by interacting with the members of the family of steroid receptor coactivators, SRC-1 and SRC-3, to regulate the transcriptional activity of genes related to cell proliferation [93].

E2 also upregulates cell proliferation in the U87GM cell line of glioblastoma. It stimulates increased activity of the transcription factor Nrf2 as well as the expression of genes involved in pathways linked to metabolism, biogenesis, and mitochondrial dynamics, which together foster an environment favorable for cancer cells [94].

The presence of E2 is reported to enhance the viability of glioma tumor cells in vitro. However, with an orthotopic model based on the implantation of U87 cells in the rat brain, contrarily, E2 administered in physiological and supra-physiological quantities (12 and 120 μg/día, respectively) was found to significantly lengthen the survival of male and ovariectomized female rats [95]. These differences show the complexity of the impact of hormones such as E2 on tumor cells.

According to preclinical studies, the E2-induced potentiation of the activity of diverse signaling pathways can lead to greater resistance to oncological treatments. It is possible to counter such resistance by administering drugs capable of interrupting the synthesis of estrogen (e.g., aromatase inhibitors, agonists of HLRH, and antagonists of ERs). The utility of these drugs should be verified with in vivo animal models and in patients. There are probably some molecular profiles that are more susceptible to endocrine therapy, which should be possible to identify with in vitro experiments [86,89,92]. Once identified, it would be recommendable to consider the molecular particularities when designing studies with in vivo animal models and selecting patients for clinical trials.

### 5.5. Agonists of ERβ

Compared to ER⍺, ERβ has been subjected to more in-depth research in relation to malignancy and has been proposed as an indicator of better prognosis in glioblastoma [96,97]. Diverse reports have shown that an elevated expression of ERβ or its stimulation with selective agonists can inhibit the processes of malignancy in gliomas by promoting the deregulation of mechanisms of DNA repair in tumors, including the pathway of homologous recombination [98,99]. The use of agonists of ERβ is another well-studied alternative, and promising results have been obtained in preclinical studies [100]. Sareddy et al. evaluated several agonists of ERβ, including a flavanone (liquiritigenin) and two synthetic compounds, diarylpropionitrile (DPN) and LY500307. In 2012, they demonstrated that liquiritigenin and DPN inhibit the proliferation of glioblastoma cell lines, and that liquiritigenin limits the growth of tumors generated by xenotransplants. The latter result stems from the capacity of the compound to diminish cell proliferation and trigger apoptosis [98]. In 2016, they reported the suppression of glioblastoma cell proliferation by LY500307, with little effect on normal astrocytes. This compound was found to promote increased cellular apoptosis by activating pathways such as p38 and JNK. It also induces cellular arrest in the G2/M phase, thus sensitizing the cells to various chemotherapy drugs (lomustine, cisplatin and temozolomide). Additionally, LY500307 was able to enhance the percentage of survival of animals through its proapoptotic activity and reduction of tumor growth. The probable mechanism of action of LY500307 is the activation of ERβ by binding directly to promoters bearing the ERE sequence or to coregulators, which allows for the recognition of the binding sites for activator protein-1 (AP-1), specificity protein (Sp-1) or NFκB in the promoters of the target genes [99].

Paterni et al. (2015) and Cao et al. (2016) investigated selective agonists of ERβ in glioblastoma models. They tried salicylaldoxime and toosendanin based on the agonist activity on ERβ. Both groups showed, in vitro and in vivo models, the efficacy in antiproliferative and proapoptotic activity. Cao et al. (2016) described the proapoptotic activity of toosendanin, mediated by an enhancement in the expression of ERβ and the presence of functional p53. According to the authors, ERβ may act as a tumor suppressor as well as a marker of sensitivity to this treatment [101,102].

## 6. Conclusions

Since tumoral heterogeneity is a major obstacle in the design of new therapies for patients with glioma, research on the types of ER isoforms present in neoplasms could be helpful in selecting the type of endocrine therapy with the greatest benefits. Preclinical studies are key to provide insights into the response to endocrine therapy (agonists or antagonists of hormone receptors) that may be expected from tumors with distinct molecular profiles. This information can then be applied to clinical studies to extend the average time of survival of patients and improve their post-treatment quality of life.

## Figures and Tables

**Figure 1 ijms-22-12404-f001:**
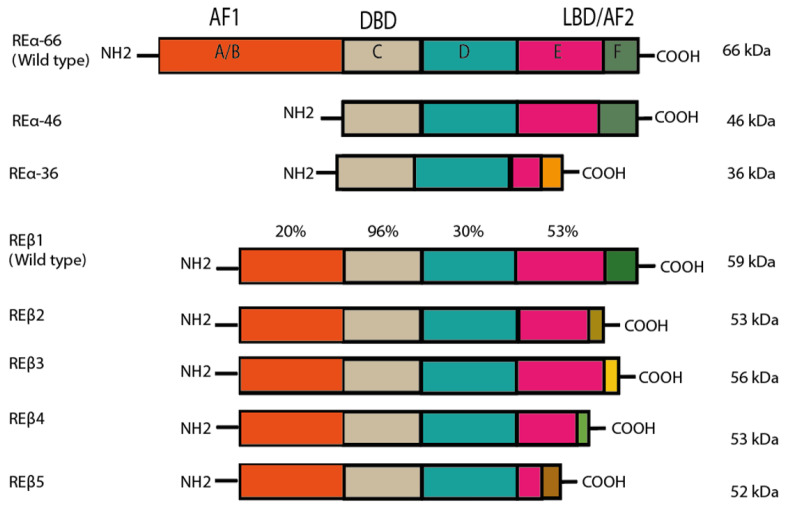
Structure of estrogen receptors (ERs), as modified by Jia et al. (2015). ER⍺ and ERβ have the same structural regions (A–F), although the degree of similarity of the domains varies. The A/B amino-terminal region contains the amino-terminal domain and the ligand-independent AF-1 domain. These domains are responsible for the recruitment of coregulators (coactivators and corepressors). The C region is the binding domain of DNA (DBD), while D is known as the hinge region. The latter also encompasses part of the ligand-dependent activation function (AF) domain and the nuclear localization signal. The C-terminus region, comprising E and F, contains the ligand binding domain (LBD) and ligand-dependent AF-2. The main isoforms of ERα are portrayed, with their respective molecular weight. A difference can be appreciated in region F of ERα36, which is caused by the transcription of exon 9. In the case of ERβ, each isoform presents variations in the F domain due to the shuffling of exon 8. Reprinted from ref. [5]. Best Pract Res Clin Endocrinol Metab, 29(4), Jia M.; Dahlman-Wright K.; Gustafsson JÅ. Estrogen receptor alpha and beta in health and disease, pp. 557–568. Copyright 2015, with permission from Elsevier.

**Figure 2 ijms-22-12404-f002:**
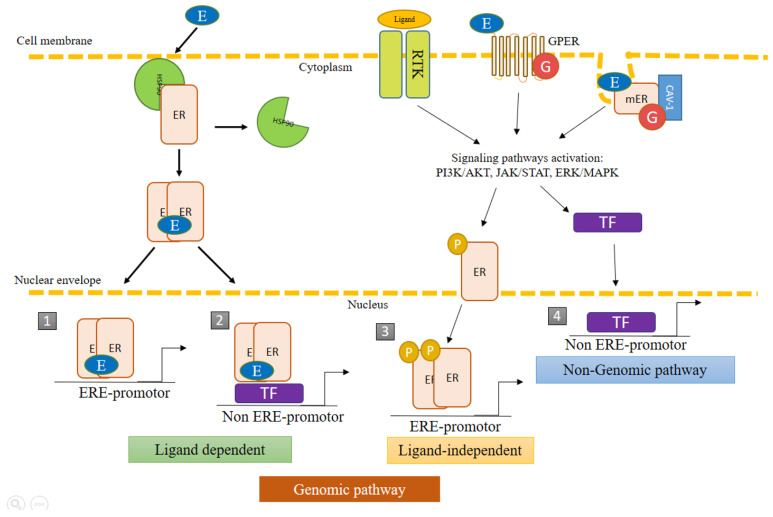
Genomic and non-genomic pathways of estrogen receptors (ERs). ERs can act through genomic pathways and nongenomic pathways. The nongenomic pathways may also result in the transcriptional activation of genes. (1) *Classic pathway*: dimerization of ERs triggered by the binding of E2, followed by nuclear translocation to regulate the transcriptional activity of genes bearing the estrogen response element (ERE). (2) *ER activity independent of ERE*: the recognition of promoters result from the protein−protein interaction of ERs with other transcription factors (TF). (3) *Ligand-independent activity*: the binding of growth factors to membrane receptors triggers protein kinase cascades that activate ERs by phosphorylation, promoting their nuclear translocation and recognition of the ERE in target genes. (4) *Nongenomic pathway*: the binding of the ligand causes mER and GPER1 to initiate signaling pathways, which promote the activation of enzymes and transcriptional factors that can regulate the transcription of diverse genes. ER, estrogen receptors; mER, membrane estrogen receptors; TF, transcription factor; ERE, estrogen response element; non-ERE, other transcription factors distinct from ERE; P, phosphorylated receptor; G, G protein-coupled receptor; GPER, G protein-coupled estrogen receptor; cav, caveolin.

## Data Availability

Not applicable.

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
