# Peer review of "Estrogen Receptors as Molecular Targets of Endocrine Therapy for Glioblastoma"

_ijms, 2021, doi:10.3390/ijms222212404_

Round 1

Reviewer 1 Report

The authors summarized the possible involvement of estrogen and its receptors in gliomas. They chose the following topics to describe in detail;
- how estrogen hormone works in the healthy organs
- how estrogen and its receptors affect the growth of the cancers
- how estrogen drugs are used for the cancers or other diseases
- how the expression of estrogen receptor can affect the therapies for glioma
- possible treatment target related to estrogen and its receptor in glioma

Basically their comments are not biased but I have several severe concerns

(1) Most of the content in the ”6 Discussion” should be stated in the section 2 to 5.

In review articles, I think, authors should review current knowledges in each topics unbiasedly. And if there are any controversial findings, authors can compare those studies and make comments to judge which one is better - those comments should be written in discussion section.

But in this article, the discussion section seemed to be just a repeat (or consecutive part) of the review topics. And because of that, especially the discussion part is quite long with a lot of duplicate statements.

2) Following contents seemed biased - only positive results were shown. 
I would like to ask the authors if there’s any controversial results.
[A] Estrogen receptor expression activate mechanisms of resistance to tamoxifen.
[B] Estrogen receptor beta agonist work as anti-GBM therapy.

Author Response

October 29, 2021

Prof. Dr. Jaroslaw Maciaczyk

Div. of Stereotactic and Functional Neurosurgery, Dept. of Neurosurgery University Hospital Bonn, Bonn, Germany

Dr. Hugo GuerreroCazares

Department of Neurological Surgery, Mayo Clinic, Jacksonville, FL, USA

Dr. Amit Sharma

Div. of Stereotactic and Functional Neurosurgery, Dept. of Neurosurgery University Hospital Bonn, Bonn, Germany

Special Issue Editors

We are resubmitting the manuscript entitled Estrogen Receptors as Molecular Targets of Endocrine Therapy for Glioblastoma” (ijms-1426775), by Andrea M. González-Mora and Patricia Garcia-Lopez

 We found the comments and suggestions by the reviewers to be extremely helpful in improving the manuscript and include a point-by-point response for the two reviewers; in the manuscript the changes are marked in blue.

 The authors declare that have reviewed and approved the final version of the manuscript for its resubmission to International Journal of Molecular Sciences and all declare that there are no conflicts of interest. The article is original, has not already been published in a journal and is not currently under consideration by any other journal. 

We look forward to hearing from you and hope that the manuscript is now suitable for publication in your prestigious journal.

Sincerely,

Patricia Garcia-Lopez. Ph.D.

Instituto Nacional de Cancerología

Av. San Fernando # 22, Col. Tlalpan, 

14000 Ciudad de México, México

Phone: (+52)(55) 36935200

e-mail: pgarcia_lopez@yahoo.com.mx   

Response to Reviewer 1

The authors summarized the possible involvement of estrogen and its receptors in gliomas. They chose the following topics to describe in detail;
- how estrogen hormone works in the healthy organs
- how estrogen and its receptors affect the growth of the cancers
- how estrogen drugs are used for the cancers or other diseases
- how the expression of estrogen receptor can affect the therapies for glioma
- possible treatment target related to estrogen and its receptor in glioma

Basically their comments are not biased but I have several severe concerns

(1) Most of the content in the ”6 Discussion” should be stated in the section 2 to 5.

In review articles, I think, authors should review current knowledges in each topic unbiasedly. And if there are any controversial findings, authors can compare those studies and make comments to judge which one is better - those comments should be written in discussion section.

But in this article, the discussion section seemed to be just a repeat (or consecutive part) of the review topics. And because of that, especially the discussion part is quite long with a lot of duplicate statements.

Response:

We found your comments and suggestions to be extremely helpful in improving the manuscript, we include the answer to your questions into the manuscript. The changes are marked in blue into the manuscript

We appreciate your comment and according to your suggestion and the suggestion of the reviewer 2, we have now incorporated all the information from the discussion into the different sections of the manuscript, discussing in each section the strengths, limitations and possible future directions of estrogen receptors as molecular targets of therapy, and we have removed the repeated paragraphs. We hope that now the information is better organized and reviewed unbiasedly. We have commented on the controversial results, which are incorporated in blue into the manuscript.

Some controversial findings with respect to ER⍺ that were included in the manuscript are the following:

Page 6, line 236-246

“Although the uncertainty about the cellular origin of gliomas and the role of ERs during the malignant transformation, relative low ER expression in tumors is clearly associated with a higher degree of malignancy. A tumor-suppressing function is frequently pro-posed for ERβ and a tumor-promoting function for ER [50-52]. 

Despite the multiple studies carried out to date, the role of ERs and E2 in gliomas is not completely clear. Hönikl et al. (2020) contradicts the previously established, by proposing that in glioblastoma, both male and female patients have an improved probability of longer survival if they exhibit an elevated expression of ER and aromatase, implying the capacity of both to suppress tumors. The authors collected tissue samples from patients and classified them according to low (<40%) and high (>40%) ER expression. Patients with high expression had a longer survival time [53]………. “

Page 7, line 256

“Unlike the findings described by Hönikl et al. (2020), the in-silico evaluation conducted by Hernández-Vega et al. (2020) revealed a lower probability of survival for patients with an elevated expression of ER and ERβ. The information for the analysis was obtained from the Cancer Genome Cancer Atlas and the GTEx database. The patients with a reduced expression of these receptors showed better survival in spite of having a robust expression of the mesenchymal subtype, considered to be the subtype with the worst prognosis in glioblastoma [55].”

Some controversial findings with respect to ER that were included in the manuscript are the following:

Page 7, line 272-315

“Concerning ERβ isoforms, it has been reported that increase in expression of isoforms ERβ2 and ERβ5 have a prognostic significance in prostate and ovarian cancer, being associated to a poor prognosis in both cancers [58,59]. However, regarding glioma, previous findings suggest that ERβ5 can play a role protective in glioblastoma. Li et al. (2013) report that ERβ5 is highly expressed in glioma compared with non-neoplastic brain tissue and this expression is increased by hypoxic conditions, promoting an inhibition of PI3K/AKT/mTOR pathway trough the induction of PTEN. Also, it was demonstrated that in U87 cell line transfected to over-express ERβ1 and ERβ5 showed a significant reduction in cellular proliferation [60].

On the other hand, Liu et al. (2018) evaluated the role of distinct isoforms in glioblastoma (e.g., ERβ1, ERβ2, ERβ4 and ERβ5) in tissues from patients as well as in cell lines and primary cultures of brain tumors. ERβ5 was more abundant in high-grade brain tumor tissue and in cell lines of glioblastoma than in tissue from healthy individuals and from low-grade glioma tumors. When examining the relation of each isoform to viability, proliferation, invasive capacity, migration, and colony-forming capacity, ERβ1 proved to have a tumor suppressor function and ERβ5 showed oncogenic properties. According to the analysis of the RNA sequence, ERβ1 modulates genes negatively correlated with the signaling pathways of NF-κB and Janus kinase signal transducer and activator of transcription proteins (JAK-STAT), while ERβ5 regulates genes positively correlated with the same signaling pathways…….. “

2) Following contents seemed biased - only positive results were shown. 
I would like to ask the authors if there’s any controversial results.

[A] Estrogen receptor expression activate mechanisms of resistance to tamoxifen.

Response:

In addition to the controversial finding mentioned before, and regarding your question of whether Estrogen receptor expression activates of resistance mechanisms of resistance to tamoxifen, we can mention that the mechanisms described in the literature are specifically related to the ER-α36 receptor. And this item is described in the manuscript as follows:

Page 7, line 266

“Although diverse studies have been carried out to determine the expression of ERs in gliomas, little evidence exists about the relevance of ER isoforms as possible therapeutic targets. Regarding ER, the ERα36 isoform is expressed in tumors from patients and in cell lines of glioblastoma, including U251 and U87-MG. Found in the cytoplasm and plasmatic membrane and anchored by caveolin-1, ERα36 has been linked to resistance to tamoxifen, perhaps mediated by a positive regulation of autophagy [56,57].”

Page 8, line 347-363

“Although tamoxifen has proven to have important activity against the progression of gliomas, there are mechanisms of resistance that can limit the efficacy of this treatment. For example, when tamoxifen binds to ER36, highly expressed in glioblastoma, the activation of this receptor contributes to the enhancement of autophagy detected in U87 and U251 cells, thus causing resistance to apoptosis. The silencing of ER36 reduces autophagy and diminishes cell viability [56].”

Increased autophagy as a mechanism of resistance to treatment is controversial. Autophagy induced by tamoxifen is described to be cytotoxic for glioblastoma (Graham et al., 2016). However, according to the report by Qu et al. (2019), tamoxifen increases autophagy and at the same time boosts the levels of ERα36 and cancer cell viability. Thereby ERα36 promotes resistance as a result of the agonist effect of tamoxifen. Furthermore, Liu et al. (2016) pointed out that the apoptosis elicited by tamoxifen through the activation of the caspase pathway and the regulation of survivin was reversed by ICI 162,486, a selective agonist of ERα36 [70,56,57].

Even though exist several in vitro assays probing the effectivity of cytotoxic tamoxifen effect using glioblastoma cell cultures, clinical trials shown discordant results in therapeutic improvement [71-76].

Page 9 line 411-421

“Even though tamoxifen is the most studied anti-estrogenic drug in clinical study phases in glioblastoma, a consensus has not been reached about the effectiveness of this treatment in high grade glioma patients, specifically in glioblastoma. According to some of the studies reported here, high doses of tamoxifen offer a positive effect in subpopulations or even promote an improvement in the mean survival of patients when combined with standard therapy. Nevertheless, other publications show a non-significant effect with respect to the control. This implies that it is necessary to consider the existing pre-clinical and clinical trials for further research with this drug to determine the appropriate experimental conditions and inclusion criteria to obtain conclusive results.

[B] Estrogen receptor beta agonist work as anti-GBM therapy

Response:

Several studies have indicated that ERβ has a different function than ERα and is considerate as tumor suppressor. It has also been reported that the expression of ERβ was decreased in high-grade tumor tissues in parallel with their loss of differentiation. ERβ expression tends to decrease with increased malignancy of the tumor; therefore, patients with ERβ-positive tumors could have a better prognosis and longer survival times. In this case, the ERβ agonists could be potential drugs with anti-glioma activities.

This topic has been discussed in the article

Page 12, line 542

“Compared to ER, ERβ has been subjected to more in-depth research in relation to malignancy and has been proposed as an indicator of better prognosis in glioblastoma [96,97]. Diverse reports have shown that an elevated expression of ERβ or its stimulation with selective agonists can inhibit the processes of malignancy in gliomas by promoting the deregulation of mechanisms of DNA repair in tumors, including the pathway of homologous recombination [98,99]. The use of agonists of ERβ is another well-studied alternative, and promising results have been obtained in pre-clinical studies [100]. Sareddy et al. evaluated several agonists of ERβ, including a flavanone (liquiritigenine) and two synthetic compounds, diarylpropionitrile (DPN) and LY500307. In 2012, they demonstrated that liquiritigenine and DPN inhibit the proliferation of glioblastoma cell lines, and that liquiritigenine limits the growth of tumors generated by xenotransplants. The latter result stems from the capacity of the compound to diminish cell proliferation and trigger apoptosis [98]. In 2016, they reported the suppression of glioblastoma cell proliferation by LY500307, with little effect on normal astrocytes. This compound was found to promote increased cellular apoptosis by activating pathways such as p38 and JNK. It also induces cellular arrest in the G2/M phase, thus sensitizing the cells to various chemotherapy drugs (lomustine, cisplatin and temozolomide). Additionally, LY500307 was able to enhance the percentage of survival of animals through its pro-apoptotic activity and reduction of tumor growth. The probable mechanism of action of LY500307 is the activation of ERβ by binding directly to promoters bearing the ERE sequence or to co-regulators, which allows for the recognition of the binding sites for activator pro-tein-1 (AP-1), specificity protein (Sp-1) or NFκB in the promoters of the target genes [99].

Paterni et al. (2015) and Cao et al. (2016) investigated selective agonists of ERβ in glioblastoma models. They tried salicylaldoxime and toosendanine based on the agonist activity on ERβ. Both groups showed, in vitro and in vivo models, the efficacy in anti-proliferative and pro-apoptotic activity. Cao et al. (2016) described the pro-apoptotic activity of toosendanine, mediated by an enhancement in the expression of ERβ and the presence of functional p53. According to the authors, ERβ may act as a tumor suppressor as well as a marker of sensitivity to this treatment [101,102].”

However, recently it has been evaluated the role of distinct isoforms of ERβ glioblastoma (Erβ1, Erβ2, Erβ3, Erβ4 and Erβ5) by Liu et al. (2018) in tissues from patients as well as in cell lines of brain tumors. ERβ5 was more abundant in high-grade brain tumor tissue and in cell lines of glioblastoma than in tissue from healthy individuals and from low-grade glioma tumors, thus ERβ5 proved to have a function oncogenic function. The ERβ5 might serve as a therapeutic target for the treatment of glioma. However, to better understand the role of ERβ5 in glioblastoma, more research is needed. This topic has been discussing in the manuscript (page 7, line 281)

Reviewer 2 Report

The manuscript reviews the topic of estrogen receptors as molecular targets of endocrine therapy for glioblastoma. Overall, the topic is of interest and the data is well presented. However, some points should be addressed and corrected.

Major comments:

The article does not adhere to the 2021 WHO classification of tumors of the central nervous system. Even in 2016 tumors were not only classified according to histopathological changes, but also according to molecular markers including IDH and LOH1p19q. In the 2021 version diagnoses such as anaplastic astrocytoma, anaplastic oligodendroglioma, and  
anaplastic oligoastrocytoma are not used anymore. Please correct the full paragraph (page 5, line 194ff). The median survival of 15 months dates back to 2005, please update according to more recent trial results (for example CORE, CENTRIC or EF-14 trial).

The authors discuss the relevance of cancer stem cells, but then talk about the expression of ERα36 isoforms in cell lines such as U251 and U87-MG.

In the first paragraph of the discussion summarizing the importance of the study quality of life is reported, but not at all in the focus of the results section.

The manuscript only mentions preclinical data. There are also a number of clinical trials on hormone therapy in glioma, for example on tamoxifen,  which should definitely be mentioned.

The discussion appears to mostly summarize data from the results section. It should also discuss strengths, limitations and possible future directions of the concept of estrogen receptors as molecular targets of endocrine therapy for glioblastoma.

Minor comments:

Some linguistic mistakes:

Examples “The hormonal factor may participate” better: hormonal factors may participate…

Line 181 “tow gonadotropins hormons”” correct to ” two gonadotropic hormones”

Page 6, line 233 „a greater the degree of malignancy” correct to “with a higher degree of malignancy”

Author Response

October 29, 2021

Prof. Dr. Jaroslaw Maciaczyk

Div. of Stereotactic and Functional Neurosurgery, Dept. of Neurosurgery University Hospital Bonn, Bonn, Germany

Dr. Hugo GuerreroCazares

Department of Neurological Surgery, Mayo Clinic, Jacksonville, FL, USA

Dr. Amit Sharma

Div. of Stereotactic and Functional Neurosurgery, Dept. of Neurosurgery University Hospital Bonn, Bonn, Germany

Special Issue Editors

We are resubmitting the manuscript entitled Estrogen Receptors as Molecular Targets of Endocrine Therapy for Glioblastoma” (ijms-1426775), by Andrea M. González-Mora and Patricia Garcia-Lopez

 We found the comments and suggestions by the reviewers to be extremely helpful in improving the manuscript and include a point-by-point response for the two reviewers; in the manuscript the changes are marked in blue.

 The authors declare that have reviewed and approved the final version of the manuscript for its resubmission to International Journal of Molecular Sciences and all declare that there are no conflicts of interest. The article is original, has not already been published in a journal and is not currently under consideration by any other journal. 

We look forward to hearing from you and hope that the manuscript is now suitable for publication in your prestigious journal.

Sincerely,

Patricia Garcia-Lopez. Ph.D.

Instituto Nacional de Cancerología

Av. San Fernando # 22, Col. Tlalpan, 

14000 Ciudad de México, México

Phone: (+52)(55) 36935200

e-mail: pgarcia_lopez@yahoo.com.mx   

Response to Reviewer 2

The manuscript reviews the topic of estrogen receptors as molecular targets of endocrine therapy for glioblastoma. Overall, the topic is of interest and the data is well presented. However, some points should be addressed and corrected.

Response:

We found your comments and suggestions to be extremely helpful in improving the manuscript, we include the answer to each of your questions into the manuscript. The changes are marked in blue into the manuscript

Major comments:

The article does not adhere to the 2021 WHO classification of tumors of the central nervous system. Even in 2016 tumors were not only classified according to histopathological changes, but also according to molecular markers including IDH and LOH1p19q. In the 2021 version diagnoses such as anaplastic astrocytoma, anaplastic oligodendroglioma, and  
anaplastic oligoastrocytoma are not used anymore. Please correct the full paragraph (page 5, line 194ff). The median survival of 15 months dates back to 2005, please update according to more recent trial results (for example CORE, CENTRIC or EF-14 trial).

Response:

We agree with your comment, according with your suggestions we have corrected the full paragraph considering the 2021 WHO classification of tumors, and we have updated the median survival according to the EF-14 trial.

Page 5, line 190.

“Glioblastoma is the most common and most aggressive primary brain tumor. In ac-cording with the fifth edition of WHO Classification of Tumors of the Central Nervous System (WHO CNS5), glioblastoma is classified using histopathological and molecular criteria. It considers histological findings, as a high proliferation rate, microvessels formation, necrosis, and the diagnostic complements with the identification of bi-omarkers as IDH-wildtype, TERT promoter mutation, +7/-10 chromosomes copy number changes and EGFR gene amplification. In the CNS5 is reported that Glio-blastoma belongs to the “Adult-type diffuse gliomas” family, with a grade 4 of ma-lignancy beings that has a poor survival and the lack of an effective therapy [38]. Based on the phase 3 EF-14 clinical trial, progression-free survival (PFS) has increased from 4.7 months to 6.7 months whit the addition of tumor-treating fields (TTFields) therapy to standard chemotherapy with temozolomide; while overall survival (OS) has improved to 20.9 months compared to 16 months with temozolomide alone, and the 5-year sur-vival rate was 13% versus 5%, [39]; unfortunately, the patients with glioblastoma have a 100% relapse rate.”

The authors discuss the relevance of cancer stem cells, but then talk about the expression of ERα36 isoforms in cell lines such as U251 and U87-MG.

Response:

We tank your comment and have separated the two paragraphs to avoid confusion.

Page 6, line 226

“When Alcantara-Llaguno et al. (2019) suppressed genes important in glioblastoma (Trp53, Pten and Nf1) in a murine transgenic model, a malignant phenotype was generated in certain types of cells, such as neural stem cells (NSC), bipotential progenitor cells, and oligodendrocyte progenitor cells. However, this effect was not observed in neural progenitor cells in the late stage of development, in newborn neurons, or in post-mitotic neurons [47,48]. Interestingly, ERα and ERβ are expressed in embryonic and adult NSC extracted from rats. A relatively high expression of ERα is found in NSC during the E15-E17 stages of embryonic development, while a reduced expression exists in adults. The expression of ERβ, on the hand, remains constant during development and is elevated in the NSC of adults [49].”

Page 7, line 266

“Although diverse studies have been carried out to determine the expression of ERs in gliomas, little evidence exists about the relevance of ER isoforms as possible therapeutic targets. Regarding ER, the ERα36 isoform is expressed in tumors from patients and in cell lines of glioblastoma, including U251 and U87-MG. Found in the cytoplasm and plasmatic membrane and anchored by caveolin-1, ERα36 has been linked to resistance to tamoxifen, perhaps mediated by a positive regulation of autophagy [56,57].”

In the first paragraph of the discussion summarizing the importance of the study quality of life is reported, but not at all in the focus of the results section.

Response:

Thank for your observation and according to your comment in the last question on the discussion appears to mostly summarize data from the results section, we have reorganized the information from the discussion within the sections of the manuscript text, and this paragraph on the quality of life of the patients has been removed.

The manuscript only mentions preclinical data. There are also a number of clinical trials on hormone therapy in glioma, for example on tamoxifen, which should definitely be mentioned.

Response:

In response to your question, we have reviewed the literature and have added the information about clinical trials on hormone therapy. We have incorporated the information in the section “5. Endocrine therapy for glioblastoma” and this information has been incorporated and classified according to the different subsections. We have included new references to support the information

As selective estrogen receptor modulators (SERMs):

Page 9, line 364-371

“In order to test the safety and efficacy of tamoxifen in the treatment of recurrent glial tumors, Couldwell et al. (1996) administered high doses of tamoxifen (200 mg /m2 for men and 160 mg / m2 for women twice daily). Among the 20 patients with glio-blastoma treated, a mean survival of 17.4 months was achieved. Previous nitrosourea therapy failed in six of the participating patients, who showed stable disease or response to treatment with Tamoxifen. Although during the study, some of the patients expe-rienced adverse effects such as deep vein thrombosis events, nausea and hot flashes. However, the author concludes that most patients tolerated the therapy well [71]”

Page 9, line 372-377

“In the same way, Chang et al. (2004) reported in a phase II study, the use of high doses of tamoxifen (escalated from 120 to 240 mg / day for one week) combined with oral etoposide (50 mg / m2 / day for three weeks) in patients with recurrent glioblastoma. The results indicate that tamoxifen has limited toxicity and was well tolerated by patients. Although efficacy was limited, probably due to highly resistant populations that prevent the synergistic effect of the combination [72].”

Page 9, line 378-386

“In another phase II study, Robins et al. (2006) reported the use of high doses of tamoxifen (80 mg / m2) orally daily divided into doses was administered after con-ventional radiotherapy (30 fractions of 2 g)……….”

Page 9, line 387-397

“In 2012, Patel et al. published a phase I study in which it was mentioned that 17 patients, recently diagnosed with a high-grade glioma, were treated with different regimens of temozolomide, radiation and tamoxifen.…..”

Page 9, line 398-410

“Moreover, in the publication of DI Cristofori et al. (2013), dense doses of TMZ (75-150 mg / m2/week with one week off), combined with daily doses of tamoxifen (80 mg / m2), were administered in patients with recurrent glioblastoma……”

As aromatase inhibitors:

Page 11, line 491

“A phase 0/1 trial is currently being conducted in recurrent high-grade patients to assess the ability of letrozole, in combination with standard therapy, to penetrate BBB and concentrate in tumoral tissue. However, results are not yet available [90]. Therefore, the usefulness of this drug in the treatment of patients cannot yet be determined”

The discussion appears to mostly summarize data from the results section. It should also discuss strengths, limitations and possible future directions of the concept of estrogen receptors as molecular targets of endocrine.

Response:

We appreciate your comment and according to your suggestion and the suggestion of the reviewer 1, we have now incorporated all the information from the discussion in the different sections of the manuscript, discussing in each sections the strengths, limitations and possible future directions of estrogen receptors as molecular targets of therapy. We hope that now the information is better organized

Minor comments:

Some linguistic mistakes:

Examples “The hormonal factor may participate” better: hormonal factors may participate…

Line 181 “tow gonadotropins hormones”” correct to ” two gonadotropic hormones”

Page 6, line 233 „a greater the degree of malignancy” correct to “with a higher degree of malignancy”

Response:

we apologize for these errors, and they have been corrected.

Page 1, line 11

Page 5, line 178

Page 6, line 238

Round 2

Reviewer 1 Report

The authors have answered to all of my previous concerns. I will recommend the editorial staffs that this article be accepted in this current form.

Reviewer 2 Report

All critical points have been improved.